# Interventions to Expand Community Pharmacists’ Scope of Practice

**DOI:** 10.3390/pharmacy12030095

**Published:** 2024-06-19

**Authors:** Zaynah Zureen Ali, Helen Skouteris, Stephanie Pirotta, Safeera Yasmeen Hussainy, Yi Ling Low, Danielle Mazza, Anisa Rojanapenkul Assifi

**Affiliations:** 1Department of General Practice, School of Public Health and Preventive Medicine, Monash University, Melbourne, VIC 3004, Australia; yi.low@monash.edu (Y.L.L.); danielle.mazza@monash.edu (D.M.); 2Health and Social Care Unit, School of Public Health and Preventive Medicine, Monash University, Melbourne, VIC 3004, Australia; helen.skouteris@monash.edu (H.S.); stephanie.pirotta1@monash.edu (S.P.); 3Pharmacy Department, Peter MacCallum Cancer Centre, Melbourne, VIC 3000, Australia; safeera.hussainy@petermac.org; 4Sir Peter MacCallum Department of Oncology, University of Melbourne, Melbourne, VIC 3000, Australia

**Keywords:** community pharmacy, RCTs, scope of practice

## Abstract

Background: The role of community pharmacists has evolved beyond the dispensing of medicines. The aim of this scoping review was to describe the interventions that expand the pharmacist’s scope of practice within a community pharmacy setting and assess their effectiveness. Methods: We performed a scoping review to identify randomised controlled trials (RCTs), published worldwide from 2013 to 2024, which focused on interventions designed to expand pharmacists’ scope of practice in the community. The review was undertaken in accordance with the Joanna Briggs Institute methodology for scoping reviews. To address the aim of this scoping review, the included RCTs were mapped to themes influenced by the Professional Practice Standards 2023 as developed by the Pharmaceutical Society of Australia: medication management, collaborative care and medication adherence. Results: Twelve studies demonstrated the potential to expand community pharmacists’ scope of practice. Two RCTs resulted in no effect of the intervention. One RCT (conducted in Italy) led to an actual change to community pharmacists’ scope of practice, with a statistically significant improvement in the proportion of patients with controlled asthma. Conclusions: On the whole, this scoping review synthesised the findings of peer-reviewed RCT studies that revealed expanding community pharmacists’ scope of practice may result in improved patient outcomes, a reduced burden for the healthcare system, and greater productivity.

## 1. Introduction

Community pharmacies are an important component of primary healthcare. Community pharmacies provide a highly accessible means of receiving medication-related advice and managing ailments, often forming the first point of contact for patients worldwide [1,2]. Community pharmacists are knowledgeable and skilled in primary healthcare delivery and present an opportunity for achieving favourable health outcomes [3]. The public perceives community pharmacists as medication experts, with a high communication skillset [4].

In recent times, the role of community pharmacists has evolved from only the dispensing of medication to becoming pivotal contributors to the management of diseases, providing patient-centred care, and promoting effectiveness-based treatments [5]. Furthermore, they can provide population-based and tailored primary healthcare services [6].

Community pharmacists have reported facilitating factors in maximising their services. These include improved cooperation between pharmacists and general practitioners (GPs), reimbursement of pharmacists, a private area within the pharmacy to facilitate patient-centred consultations, up-skilling of staff, and external support for them [7]. Conversely, the barriers that hinder the role of community pharmacists include low consumer awareness of existing programs [8], lack of time [9], resources and self-confidence [10], and GPs not being engaged in the process [11].

To address the known limitations of the role of community pharmacists, expanding their scope of practice may optimise the overall patient healthcare outcomes and reduce the burden of disease across communities, particularly those in rural and regional areas, given that access to health professionals is lower in these regions when compared to metropolitan areas [12]. Expansion of the provided services may also broaden the knowledge, skills, and expertise of pharmacists when delivering new health services [12].

The concept of expanding community pharmacists’ scope of practice differs worldwide. The terms expanded and extended scope of practice, though used interchangeably in the literature, can define a difference in terms of whether a service goes beyond or fits within the professional scope of practice. For the purposes of this paper, we use the term expanded scope of practice. For example, in Canada, expanding community pharmacists’ scope of practice includes hospital and community pharmacists being able to renew, refuse to fill, adjust or substitute prescriptions, as well as to initiate drug therapy for certain self-limiting conditions, administer injections and vaccines, and interpret and order laboratory tests [13]. To supplement these examples, the Canadian Pharmacists Association has defined the expanded scope of community pharmacy practice as a means “*to adapt pharmacy practice and programs for better health and value for payers, pharmacists, other health care providers, and all Canadians, with a particular emphasis on smoking cessation; influenza vaccination administration; dyslipidaemia; diabetes; hypertension and respiratory conditions*” [14]. Comparatively, in the Australian context, the National Competency Standards Framework for Pharmacists states that “*the boundaries of professional pharmacy practice, the healthcare needs of patients and the applicable workplace policies, as scope of practice*” [15]. Some growing examples of the expanded scope of community pharmacy practice in Australia include (but are not limited to) the implementation of a urinary tract infection program to supply indicated antibiotics, the delivery of all the vaccines under the National Immunisation Program (NIP), and the continued supply of oral contraceptive pills [16]. In Australia, the community pharmacy scope of practice is defined as “*a time-sensitive, dynamic aspect of practice which indicates the professional activities that a pharmacist is educated about, competent and authorised to perform, and for which they are accountable*” (p. 5) [3,15]. Therefore, expanding the community pharmacy scope of practice refers to pharmacists carrying out their roles and responsibilities beyond the current standard of practice.

Several systematic literature reviews and two Cochrane Reviews that have included studies on expanding the scope of community pharmacy practice have also been published in the last 10 years. These reviews/studies have focused on a single medical health issue, such as osteoporosis [17], or a single treatment or management plan, such as opioid therapy [18], asthma management [19], lifestyle health management [20], medication review with the GP and pharmacy-based interventions in improving depression outcomes in adults [21].

Despite the literature base to date, systematic and rigorous research has yet to be conducted that moves beyond: (1) pharmacist views and experiences; (2) an evaluation of current pharmacist interventions that are within their regulatory control; and (3) a focus on single medical issues, treatments and/or management plans. To inform further expansion of the community pharmacists’ scope of practice, a more integrated and holistic understanding is needed of which interventions have successfully resulted in an expanded scope of practice and improved outcomes for patients. Hence, our goal was to identify and map the current emerging evidence of interventions that expand the pharmacist’s scope of practice within a community pharmacy setting by conducting a scoping review. The objective was to describe the interventions that expand the pharmacist’s scope of practice within a community pharmacy setting and assess where there has been a change to the scope of pharmacy practice by those interventions and to determine their effectiveness. Exploring the current scope of practice in community pharmacy and the means of expanding it will help policymakers understand its impact on financial and health outcomes, and it will contribute to future recommendations for pharmacy practice.

## 2. Materials and Methods

The method outlined here followed the Scoping Review Framework as guided by the Joanna Briggs Institute (JBI) Reviewer’s Manual [22]. The JBI framework promotes the effective and rigorous synthesis of material related to a topic area by: (1) identifying the aim and objectives (noted above); (2) searching for relevant studies; (3) selecting included studies; (4) charting the data; and (5) collating and summarising the findings [22]. The JBI critical appraisal tool [23] was used to provide the systematic, transparent and trustworthy appraisal of resources for the research synthesis within this scoping review to determine the quality of the evidence included [22].

### 2.1. Inclusion and Exclusion Criteria

To be included in this review, the retrieved studies needed to meet the following inclusion criteria: (1) all the studies must be an RCT; (2) conducted in a community pharmacy; (3) written in the English language; (4) the RCT must have been completed no earlier than January 2013 and no later than May 2024; and (5) identified an actual or potential change to community pharmacists’ scope of practice and the assessment of the effectiveness of these interventions. Articles that reported observational studies, commentaries, conference abstracts, case studies, opinion pieces, reports, grey literature, experiments, interviews, narrative reviews, pilot studies, scoping reviews and protocol papers were excluded from this review, as well as RCTs not published in the English language. No restriction to region, country, or geographical area was considered for RCTs facilitated in community pharmacy. Studies where patients transitioned from hospital to community pharmacy to receive the intervention were also included to demonstrate the expanded scope of community pharmacy practice.

### 2.2. Search Strategy

The search strategy was developed with the assistance of a team of medical librarians at Monash University, with consideration of the goal and objective, PICO (population, intervention, comparator, outcome), and inclusion and exclusion criteria. It included various combinations and Medical Subject Heading (MeSH) terms for the following key concepts: pharmacy, interventions, trials, and scope of practice. The search strategy also included thesaurus and keyword searches and was refined as the authors became familiar with the literature and resources. This refinement included removing the terms “community networks”, “community health services” and “placebo”, as this broadened the depth of the studies beyond the scope of the review and, thus, may not have maintained the specificity required. The term “randomized” was incorporated into the search strategy, as well as the British English spelling “randomised”, to ensure that studies from the United Kingdom, Canada, Australia, and other countries that use British English were captured in the review. The term “drug store” was also included in the search strategy as this is a phrase commonly used in American literature to identify the community pharmacy setting. The term “implement” adjacent to “practice” was adapted in the search strategy to ensure that a full breadth of RCTs were captured in this scoping review. The following databases were searched to inform the scoping review: Medline (ALL), Web of Science, Scopus, CINAHL Complete, Cochrane Library, Embase (OVID) and Emcare (OVID). A date range of the previous 11 years (for the recency of the evidence) from January 2013 to May 2024 was incorporated into the search.

The search strategy was pre-tested on a set of five test studies to ensure the robustness of the strategy. When the search strategy was implemented, four of the five (80%) studies were returned in the same search, verifying the reliability of the search strategy. The detailed search strategy used for all the databases in this scoping review is included in Appendix A. The final search results were exported to EndNote 20 and then to Covidence, a systematic review management tool, to assist with removing duplicates and screening the studies. Title, abstract and full text screening was completed by study authors Z.Z.A. and Y.L.L. based on the study selection and inclusion criteria, with all the minor discrepancies between investigators resolved by discussion until a consensus was reached. When a conflict remained because a consensus could not be reached, a mediator (S.P.) was consulted to help resolve this. Studies where the full text was not available were excluded.

### 2.3. Extraction of Results and Data Synthesis

The PRISMA extension for the scoping review framework (PRISMA-ScR) was used to support the search and data extraction process [24]. Z.Z.A. completed the data extraction for the studies included and regularly discussed the results with the team. Y.L.L. completed a 10% check of the data extraction.

The data-charting process was vigorous and extensive, and it included extracting, analysing, and presenting evidence from each study to address the study aim and objective (see Appendix A). To address the aim of this scoping review, the included RCTs were also mapped to the Pharmaceutical Society of Australia’s *Professional Practice Standards 2023* themes of [25] medication management, collaborative care, and medication adherence (see Appendix A). This was performed to ensure that the findings reported were contemporary to how pharmacists currently practise, making this review transferable and applicable in expanding community pharmacists’ scope of practice [25].

### 2.4. PSA Professional Practice Standards 2023

The Pharmaceutical Society of Australia (PSA) is the only peak national professional pharmacy organisation that is recognised by the Australian government to represent all of Australia’s 36,000 pharmacists working in all sectors nationwide [25]. The PSA guides the innovative and evidence-based healthcare service delivery by all pharmacists across all sectors. The PSA governs high-quality pharmacist development and practice support, and it serves to be the custodian of the professional practice standards and guidelines that house all quality and integrity practices in pharmacy [25]. Noting that the primary author of this scoping review is an Australian-trained pharmacist, the PSA’s Professional Practice Standards 2023 [25] have been used to map the outcomes of this scoping review. This is to contextualise the global evidence from the scoping review to the Australian setting, as a way to highlight how Australia can achieve such interventions implemented globally and introduce these methods to the Australian way of practice. Australia is currently undergoing a scope of practice review to evaluate the barriers and incentives for primary healthcare professionals working to their full scope of practice [26]. Hence, the current review addresses key international lessons that Australia can implement in their efforts to ensure safe and affordable care for patients.

The three themes of “medication adherence”, “medication management”, and collaborative care all relate to the PPS and therefore to the expanded scope of community pharmacy practice as a symbiotic relationship [25]. These three themes are mentioned frequently in the PPS to highlight their importance when carrying out the role of a pharmacist.

The PPS are developed in such a way as to reflect the dynamic healthcare environment pharmacists work within. The safe and effective use of medicines, including medication adherence, is at the core of pharmacy practice, and this is depicted at all the phases of the medicines management cycle (MMC) [25]. To demonstrate this change, the PPS have been redesigned to reflect the alignment with the MMC. The Professional Practice Standards’ medicines management cycle encompasses “review and monitoring”, “prescribing”, “dispensing and preparation”, and “administration” [25]. These counterparts feed into each other to cohesively form the medicines management cycle.

Medication adherence and medication management form important parts of the decision-making process from the point of the medication being prescribed, evaluation of the therapeutic appropriateness of treatment, and then the decision to dispense or refuse supply of the prescription item and/or escalate a recommendation to the treating team. These principles are used in this scoping review to demonstrate that an expanded scope of community pharmacy practice that involves medication adherence and medication management may improve the patient-related health outcomes and therefore may reduce the burden on the healthcare system. It illustrates that medication management follows a cyclical nature and that all the parts work in cohesion to assist the pharmacist in their decision-making processes. Collaborative care is defined by the PPS as a fundamental standard of a pharmacist’s practice [25]. The PPS define the concept of collaborative care as that “*the pharmacist collaborates with other members of the healthcare team to deliver coordinated, person-centred care to improve health outcomes and optimise the quality use of medicines*” ([25], p. 17). Collaborative care in conjunction with medication adherence and medication management contribute to an effective expanded scope of community pharmacy practice.

Pharmacists continue to play a crucial role in ensuring safe and reliable access to medicines and healthcare services, especially during public health emergencies such as floods, fires, and pandemics [25]. As the scope of community pharmacy practice continues to expand, it is of utmost importance that pharmacists practice in a way that is up-to-date with the current standards and therefore evidence-based to guide how they go about facilitating their role [25].

## 3. Results

### 3.1. Overview

The initial search identified 9935 citations. After screening the titles and abstracts, 249 studies proceeded to full-text screening. The full texts of the 249 studies were assessed for eligibility, where a further 235 were excluded with reasons outlined in the PRISMA-ScR diagram (Appendix A). There was complete (100%) agreement for the included RCTs based on the screening process that took place using Covidence and the RCTs being re-checked by two researchers for inclusion in the scoping review. This resulted in 14 studies being included.

### 3.2. Critical Appraisal Tool

Using the JBI Critical Appraisal Tools [23], Z. Z. A. critically appraised all fourteen RCTs to test the validity and rigour of the included studies; Y.L.L. critically appraised three RCTs and total agreement was achieved in relation to the quality standards for inclusion. A threshold of 70% agreement was considered to be an acceptable level for this type of research.

### 3.3. Characteristics of Included Studies

Appendix A summarises each paper included in this review and reports the authors, year, and study characteristics. Detailed information about the community pharmacy sites for each of the included papers is included in Appendix A. The included RCT studies were published between 2015 and 2022; four studies were from Canada [27,28,29,30], and one each from the USA [31], Egypt [32], Qatar [33], Italy [34], Iran [35], Croatia [36], Netherlands [37], UK [38], Australia [39] and Jordan [40]. One RCT was carried out in a “hybrid-setting” where the participant was discharged from hospital and received the intervention (or part of the intervention) in a community pharmacy [31]. The efficacy of the intervention and change to the community pharmacist’s scope of practice were the key foci of this scoping review.

### 3.4. Outcomes

Twelve studies demonstrated the potential to expand community pharmacists’ scope of practice. Two RCTs resulted in no effect of the intervention being found [35,37].

The outcomes of the included studies (Appendix A) were aligned with the three themed categories/topic areas outlined in the *Professional Practice Standards 2023* [25]: medical management; collaborative care; and medication adherence, as outlined by the Pharmaceutical Society of Australia.

#### 3.4.1. Medication Management (n = 5)

Five RCTs in this scoping review focused on whether medication management can be improved through an expanded community pharmacy scope of practice [27,34,35,36,39]. The RCTs were conducted in Canada [27], Italy [34], Iran [35], Australia [39] and Croatia [36]. The practical expanded scope of practice that took place relating to medication management includes community pharmacists monitoring patient progress through weekly comprehensive visits for 12 weeks [27], facilitation of a medication review by a community pharmacist [34,35], provision of a follow-up plan, pill-box supply, frequent consultation with physicians to track international normalised ratio (INR) and warfarin management [36], and diagnostic screening of diabetes [39]. The pharmacists demonstrated improvements in medication management by promoting clinically meaningful improvements in haemoglobin A1C, anthropometrics, blood pressure, and triglycerides (all *p* < 0.0001) [27], and increasing the time in the therapeutic range (TTR) for warfarin to optimise the therapy (such that 93% vs. 31.2% for intervention and control, respectively; *p* < 0.001) [36].

A medicine use review (I-MUR) intervention for patients with asthma in Italy [34] resulted in a 35.4% improvement in the medication adherence rate 3 months post-intervention and 40.0% at 6 months (*p* < 0.01). This intervention, which consisted of a systematic, structured face-to-face consultation with a pharmacist covering the asthma symptoms, medicines used, attitude towards medicines and adherence, and the recording of pharmacist-identified pharmaceutical care issues was also more cost-effective than the usual care at nine months post-intervention [34]. The success of this intervention was so impactful that it became the first community pharmacy service to be implemented in Italy, where accredited pharmacists can undertake a structured adherence-centred review with patients using multiple medicines, especially those receiving medicines for asthma and other long-term conditions [34].

A novel study implemented in Australian community pharmacies [39] introduced a point-of-care (POC) test adjunct to the validated 10-item Australian Type 2 Diabetes Risk Assessment (AUSDRISK) tool. The POC test following the AUSDRISK screening was either a random or fasting small capillary blood glucose test (scBGT), or a glycated haemoglobin (HbA1_c_) for those at an elevated risk [39]. The addition of the scBGT and HbA1_c_ POC tests to the AUSDRISK assessment increased the uptake of community pharmacist referrals for diagnostic testing and, in turn, improved the rate of newly diagnosed type 2 diabetes cases compared with the AUSDRISK alone [39]. Of the total study population that was screened, the rates of T2DM diagnoses were significantly higher in the HbA1_c_-POC group (1.5%) compared with the AUSDRISK-alone/control group (<0.8%) and the scBGT-POC group (<0.6%) [39]. Thus, in the Australian community pharmacy context, the most effective method to reveal undiagnosed T2DM was a stepwise approach using initial risk assessment (via implementation of the AUSDRISK validated tool) and, if warranted, an HbA1_c_-POC test and referral [39].

In contrast, an RCT carried out in Canada [35] that focused on a community pharmacist-based diabetes education program did not result in statistically significant improvements in the amount of A1C reduction in the intervention group.

#### 3.4.2. Collaborative Care (n = 4)

Four RCTs demonstrated an expanded community pharmacy scope of practice by focusing on collaborative care [31,32,38,40], with one of these exploring pharmacists’ interventions at the transition of care from hospital to community pharmacy [31]. The RCTs were conducted in the USA [31], Egypt [32], the UK [38] and Jordan [40]. The practical expanded scope of practice relating to collaborative care includes community pharmacists enhancing the transition of care by assisting with the hospital discharge process for the safe supply and reconciliation of medications [31], the implementation of a health coaching model [32,40] and the implementation of a bridging-intervention to reduce the risk of unintended pregnancy [38]. One study [31] demonstrated that for the primary outcome measure of a 30-day readmission rate, community pharmacist interventions led to a significant difference between the intervention and control groups (1.6% vs. 10.7%; *p* = 0.02) due to implementing discharge medication programs and improving the overall medication adherence [31]. This means that without the intervention, patients were 10 times more likely to be readmitted to hospital [31].

Another study aimed to reduce the rate of unintended pregnancies in the UK by offering women in a community pharmacy setting who requested the emergency contraceptive pill (ECP) a 3-month supply of the progestogen-only pill (75 microgram desogestrel) in conjunction with a rapid access card for a participating sexual and reproductive health clinic [38]. Post-implementation, the proportion of women using effective contraception was 20.1% greater in the intervention group (mean 58.4%, 48.6–68.2) than compared to the control group (mean 40.5%, 29.7–51.3) [38].

Two studies utilised a coaching model to improve patient outcomes [32,40]. The first study involved a pharmacist-led virtual coaching program delivered via Zoom^®^ over a month, with a two-week follow-up period. Following program completion, the proportion of behaviours, such as disinfecting surfaces, not touching the T-zone, and avoiding sharing personal belongings with colleagues at work, significantly increased by 27.89%, 24.49% and 40.82%, respectively (all with *p* < 0.05) [40]. In the second study, significant improvements in the proportion of high physical activity levels (13.91% at baseline and 53.04% three months after coaching), practising a healthy diet (26.95% at baseline and 62.60% three months after coaching) and completing a breast self-exam three months post-program completion was observed.

#### 3.4.3. Medication Adherence (n = 5)

Five RCTs reported on medication adherence and how it can be improved by expanding the scope of practice of community pharmacists [28,29,30,33,37]. The RCTs were conducted in Canada [28,29,30], Qatar [33] and the Netherlands [37]. The practical expanded scope of practice that took place relating to medication management includes community pharmacists participating in a telephone-specific service for the follow-up of patients [33] and pharmacist-prescribing services [28,29,30]. One of the studies [37] focused on improving medication adherence through a telephone-service but did not improve the patient outcomes for patients initiating antidepressants. In contrast, in four of the five studies [28,29,30,33], pharmacists were able to demonstrate improvements in medication adherence.

In one study [33], pharmacists received training and role-played examples of how to provide thorough and comprehensive antibiotic counselling to patients. The pharmacists in the intervention arm contacted the participants three to five days after dispensing the antibiotic. This phone call allowed the pharmacists to interview the patients regarding antibiotic adherence, side effects, and symptom relief, as well as assisting in improving compliance, and to re-emphasise the importance of completing the whole course of antibiotics as prescribed [33]. The pharmacists also referred the patients back to their prescriber if their symptoms were not being resolved. The antibiotic adherence rates did not differ between the groups of standard care, counselling, and call-back [33].

An RCT conducted in the Netherlands involved patients receiving telephone counselling 7–21 days after newly starting an antidepressant, bisphosphonate, Renin-Angiotensin System (RAS) inhibitor, or statin, as part of the community-pharmacy-based intervention [37]. In this study, it was found that adherence was significantly higher in patients newly starting an RAS inhibitor (84.1% compared to 78.5% in the control-group), statins (80.5% compared to 75.1% in the control-group), and bisphosphonates (72.2% compared to 73.3% in the control-group) compared to patients newly starting antidepressants but not benefitting from the intervention [37]. No significant difference in the adherence rates were eventuated between patients in the intervention arm (62.7%) and patients with usual care (66.8%) [37]. Of those participants newly starting therapy with an antidepressant, 47.5% of participants discontinued therapy in the intervention arm compared to 42.7% of participants in the usual care arm [37]. It is possible that the significant difference in dropout rates between the intervention (47.5%) and the control groups (42.7%) contributed to educational interventions (such as telephone counselling) alone not being enough and that complex interventions are needed [37]. Additional reasons for this may be that those using antidepressants may be more likely to experience adverse drug events [37], such as weight gain, reduced libido and increased risk of drug interactions [37].

In a series of three RCTs conducted by the same lead researcher (Tsuyuki et al.) [28,29,30], the RxACT study [29] focused on maintaining optimal cholesterol targets in patients, the RxEACH study evaluated the effect of a pharmacist case-finding approach and intervention program on the estimated cardiovascular (CV) risk in patients with diabetes [30], whilst the aim of the RxACTION study [28] was to optimise the blood pressure outcomes. In the RxACT study, the intervention involved participant identification, assessment, care plan development, education/counselling, prescribing/titration of lipid-lowering medications and close follow-up and monitoring after the intervention was implemented [29]. The RxACT study resulted in 25% more patients achieving the guideline targets for low-density lipoprotein-_c_ (LDL-_c_) levels and reaching their LDL cholesterol goal compared to usual care [29]. In the RxACTION study with a similar intervention but different population target (of those with above-target blood pressure), the intervention group had a mean ± SE reduction in systolic blood pressure (BP) at 6 months of 18.3 ± 1.2 compared with 11.8 ± 1.9 mm Hg in the control group; an adjusted difference of 6.6 ± 1.9 mm Hg (*p* = 0.0006) [28]. The adjusted odds of patients achieving the BP targets was 2.32 (95% confidence interval, 1.17–4.15 in favour of the intervention) [28]. The R_x_EACH study also implemented patient assessment, laboratory testing and assessment, individualised CV risk assessment, and, where clinically appropriate, treatment recommendations, involving prescription adaptations and/or initiation to fulfill glycaemic, blood pressure and lipid-control targets and tobacco cessation [30]. The estimated CV risk was reduced from 26.9+/−21% to 26.5+/−21.3% in the control group and from 25.8+/−19.4% to 20.1+/−17.2% in the intervention group within the 3-month follow-up period (ARR 5.38; 95% CI 4.24–6.52; *p* < 0.001) [30].

## 4. Discussion

Across countries globally, community pharmacy services have increased in diversity, complexity, and type to cater to different healthcare systems and a wide range of medical conditions. This shift in service provision requires an expanded scope of community pharmacy practice and an emphasis on medication management, collaborative care and medication adherence [25] to reflect the changes in pharmacy practice. To the best of our knowledge, the scoping review we conducted is the first of its kind to focus on a broad range of interventions that expand the pharmacist’s scope of practice within a community pharmacy, moving beyond a single medical issue and treatment/management plan and pharmacist views and experiences. Our aim was to describe these interventions and determine their effectiveness, which was met.

This scoping review identified that when community pharmacists are supported to practice beyond their standard scope of care, it has the potential to improve positive patient outcomes. With community pharmacists intervening more often, the included studies showed that their health-related advice and care resulted in improved medication management and medication adherence [30], improved care transitions [31], increased uptake of effective contraception (e.g., long-acting reversible contraception) [38], and better management of chronic conditions, including (but not limited to) asthma [34] T2DM [27], high blood-pressure [28] and dyslipidaemia [29]. The ways in which pharmacists were supported to deliver additional services were also important components of the intervention delivery, as identified through this scoping review. This includes training modules and continuing professional development [38], case-scenario simulations [37], educational tools [27], and coaching by dedicated experts in the field [29].

The 14 included studies were also grouped according to themes influenced by the *Professional Practice Standards 2023*, as developed by the Pharmaceutical Society of Australia: *medication management [27,34,35,36,39], collaborative care [31,32,38,40] and medication adherence [28,29,30,33,37]*. Five [27,34,35,36,39], four [31,32,38,40], and five [28,29,30,33,37] of the studies focused on whether medication management, collaborative care, and medication adherence can be improved through expanded community pharmacy practice, respectively. The findings of all the RCTs revealed significant improvements except in two RCTs that focused on medication management [35] and medication adherence [37].

Of the twelve successful trials, only one intervention led to an actual change to the community pharmacists’ scope of practice [34]. This study resulted in a statistically significant improvement in the proportion of patients with controlled asthma [34]. The intervention (known as the “Italian-Medicines Use Review/I-MUR”) was also shown to be cost-effective. The Italian Government and Ministry of Health have since changed the scope of community pharmacy practice, with the I-MUR being the first nationally funded pharmaceutical service in Italy [34]. This has revolutionised community pharmacy practice to promote the change from a mainly product-based model of care to a more patient-centred and clinically oriented role on the part of the community pharmacist.

Italy is now considering expanding the service to other respiratory conditions and could enable community pharmacists to support the care of patients with a wide range of long-term conditions in the future [34]. At the time of this study, community pharmacists in Italy were involved in little clinical input, being mainly a supply function. The patients were, therefore, not accustomed to a pharmacist taking an interest in their clinical status and providing information about optimising medicine use. This change in pharmacy practice may have contributed to the effectiveness of the intervention and therefore signifies the positive impact a change in the community pharmacy scope of practice can provide in achieving positive healthcare outcomes [34].

In a practical setting, the 12 successful studies, and in particular the I-MUR study, highlight that regular contact with the pharmacist is an important factor in improving outcomes for patients. The findings indicate the growing need for community pharmacy services delivered to patients to exhibit an expanded scope of practice. This is because there is an obvious correlation between pharmacists being able to practise to their fullest ability and the subsequent positive healthcare outcomes for the patient [41] that can be achieved. However, the scope of practice that community pharmacists can practise within is directly impacted by multiple levels of influence, involving individual, interpersonal, institutional, and public policy-related factors [41]. Most RCTs included in this scoping review saw high patient acceptance and satisfaction with an expanded scope of community pharmacy practice, suggesting that increasing the roles and responsibilities of pharmacists is greatly supported [41].

Importantly, this scoping review considered the possibility of pharmacist prescribing, which would be a significant change to the pharmacists’ scope of practice [42]. Non-medical prescribers, particularly pharmacists, can achieve comparable clinical outcomes with doctors for certain health conditions [42]. It is important to note, however, as this scoping review included RCTs that had been completed in 2013 to 2024, that pharmacist prescribing (PP) at that time point was still a developing area worldwide. Recently, legislation supporting PP has been implemented in the UK, Canada and New Zealand, but not yet in Australia [42].

Three RCTs based in Canada considered pharmacist prescribing as part of their role [28,29,30] in the context of dyslipidaemia management [29], blood-pressure management [28] and diabetes [30]. Whilst these studies are yet to be implemented in real-life practice in Australia, they nonetheless demonstrate that pharmacist prescribing promotes a clinically significant increase in the proportion of patients achieving the guideline treatment targets [28,29,30] compared with usual care. Given the accessibility of community pharmacists to patients who have a high burden of illness from most comorbidities, pharmacist prescribing could have an important impact on public health [28,29,30]. Pharmacist prescribing may overcome the ceiling effect that may be associated with recommendation-based care [28].

In addition, the Bridge-It trial highlighted the effectiveness of community pharmacists supplying the progestogen-only pill (75 microg desogestrel) and rapid access to a participating sexual and reproductive health clinic when women are seeking the emergency contraceptive pill (ECP), which then subsequently increased the uptake of effective contraception and thereby lowered the risk of unintended pregnancy [38]. This reduces the burden on the primary healthcare system but also provides an equitable and accessible mechanism for women to explore their reproductive health and maintain autonomy in doing so. Interestingly, community pharmacists in the UK have had jurisdictive power from 1 December 2023 to supply the oral contraceptive pill (OCP) without needing a prescription from a GP [43]. This is similar to practices established in Canada and New Zealand. Australia has recently engaged in implementing trials relating to the resupply of OCPs, and so it is evident that the community pharmacy scope of practice in the area of reproductive health is rapidly expanding [26].

The findings of this scoping review also recognised that the pharmacist call-back service is a simple and inexpensive intervention that can effectively capture opportunities for improving appropriate antibiotic use, particularly when related to medication adherence. Health coaching, which includes the integration of health technology such as telepharmacy, is a newfound mechanism implemented by pharmacists, showing positive results in patient-centred education and counselling, patient compliance, accountability, and improving long-term conditions, in conjunction to cost benefits [40]. However, the employment of telepharmacy in pharmacy practice is still in its beginnings [40]. Recent evidence has highlighted that routine follow-up of people with chronic disease is not happening enough in community pharmacy [44]. A major barrier to the routine monitoring and follow-up of patients in community pharmacy is the poor understanding of the scopes of practice/services of pharmacy professionals in the healthcare system [44].

Pharmacists can make substantial contributions to innovative, collaborative, interdisciplinary practice models to improve patient care [31]. The care transition is an emerging space in pharmacy, which is defined as a time-limited service to ensure the continuity of care and to avoid poor health outcomes while a patient is transitioning from one setting of care to another [31]. This commonly involves predischarge interventions (e.g., patient education, discharge planning, medication reconciliation, scheduling a follow-up appointment), which is a hospital-driven process; postdischarge interventions (e.g., follow-up telephone call, communication with ambulatory provider, home visits); and bridging interventions (e.g., transition coaches, patient-centred discharge instructions, clinician continuity between inpatient, and outpatient settings) [31]. Community pharmacists are well placed to bridge transitions by receiving patients from hospital and supporting medication management and monitoring at this time-critical juncture, where the patient often experiences a medication change and treatment rationalisation [31]. The transmission of patient information from the inpatient setting to the community pharmacy will be key to reducing fragmentation in healthcare [31]. This intervention demonstrating an expanded scope of community pharmacy practice has been shown to reduce hospital admission rates ten-fold [31], and it is an opportunity for community pharmacists to engage with patients regarding ongoing care, monitoring, resolution of pharmaceutical-related problems, and harmonising health [31].

Unlike the situation in high-income countries such as Australia, New Zealand, Canada, the UK and the USA, to accommodate the lack of community pharmacists and the low presence of community pharmacies outside big cities in Ghana (such as the Greater Accra region), the Pharmacy Council of Ghana authorised the opening of chemical shops by people who were not pharmacists, that is, licensed chemical sellers (LCSs) [45]. To be qualified as an LCS, a person must at a minimum pass secondary school-level education, with basic knowledge of healthcare [46]. Pre-licensing training is mandatory, and this license is renewed yearly [46]. The LCSs are authorised to retail over-the-counter medicines to communities that the Pharmacy Council considers poor [46]. These medicines include analgesics and antimalarials, and the only antibiotic allowed is co-trimoxazole [46]. Community shops form the bulk of the basic medicine supply in Ghana because of lacking the resources, funding, education, and facilities to have more community pharmacists than LCSs [46]. This highlights a major gap in healthcare when considering low-income countries but demonstrates the measures necessary for a variant pharmacy service to be able to service the community in Ghana, even if they are not community pharmacists. To support the expanded scope of pharmacy practice in Ghana, a medication management intervention for malaria involved the use of a rapid diagnostic test before dispensing any antimalarial medication [46]. Whilst there was some prescribing of antimalarials for test-negative cases, clients visiting chemical shops in the intervention arm with complaints of fever were three times more likely to receive appropriate treatment for malaria fever than those visiting chemical shops where no rapid diagnostic testing was being performed [46]. The findings of this study showed that although there is a shortage of community pharmacists in Ghana, this does not have to be a barrier to accessing robust healthcare through the expansion of the scope of practice of LCSs.

For community pharmacists to provide the effective and successful delivery of the expanded scope of community pharmacy practice, it is recommended that: (1) evidence-based guidelines or protocols are implemented to assist with assessment and diagnosis; (2) the pharmacist comprehensively documents the patient interaction and integration of care via collaboration with GPs and other suitable health services as well as digital health technology, referrals, monitoring, and follow-up; (3) a private consultation room is available in the pharmacy to maintain patient privacy and confidentiality; and (4) there is greater public awareness of pharmacy services and what they do aside from dispensing of medicines [41]. These recommendations will not be possible without the drive from peak professional pharmacy bodies and the government to legislate the expanded scope of community pharmacy practice [41].

### 4.1. Strengths

The notable strengths of this scoping review are that a critical appraisal of the included studies was undertaken, where 100% agreement was achieved between reviewers. This ensures that the findings drawn from this scoping review are reliable and transferable.

As this scoping review considers evidence from the last 11 years, this helps to filter the content to RCTs that are relevant in practice and reflect the changes to legislation and pharmacy practice standards that govern the way a pharmacist carries out their role.

Additionally, when a review of the literature was undertaken, most systematic literature reviews relating to an expanded scope of community pharmacy practice were restricted to the one discipline, such as warfarin management [36], type 2 diabetes mellitus management [27] or blood-pressure management [28]. This scoping review has successfully provided a cross-section of multiple disciplines in pharmacy practice so that consideration can be given to all types of healthcare and all types of medical conditions.

### 4.2. Limitations

Despite its contribution to the knowledge of expanding community pharmacists’ scope of practice, this scoping review has some limitations. Some relevant studies and reviews may have been overlooked, since only English language articles were included. This may also mean that the included articles may not be complete, as some Canadian provinces use French as their official language. Furthermore, some of the studies did not meet the specified sample size and so the results may not be applicable, irrespective of the outcome of the RCT. Nevertheless, this review highlighted the obvious benefit of expanding community pharmacists’ scope of practice and the need for pharmacists to be able to practice to their full extent to advocate for positive patient outcomes.

## 5. Conclusions

On the whole, this scoping review synthesised the findings of peer-reviewed RCT studies that revealed expanding community pharmacists’ scope of practice may result in improved patient outcomes, a reduced burden on the healthcare system, and greater productivity. To ensure an expanded scope of community pharmacy practice, more research and advocacy is needed, including reforming relevant policies and guidelines. This will help to address the common barriers community pharmacists often rate highly as deterrents to practice, including a lack of compensation [41], time constraints [41], low acceptability from other healthcare professionals in collaborative opportunities [41], and a lack of training opportunities [41]. By expanding community pharmacists’ scope of practice, this may reduce the cost burden on the healthcare system because pharmacists will be better utilised in their roles [41]. Expanding community pharmacists’ scope of practice may also be applicable in low-income countries such as Ghana, since the majority of the medication supply is controlled by LCSs who have the potential to expand their scope of practice, given their prevalence and higher accessibility than community pharmacies [46].

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
