# Peer review of "Interventions to Expand Community Pharmacists’ Scope of Practice"

_pharmacy, 2024, doi:10.3390/pharmacy12030095_

Round 1

Reviewer 1 Report

Comments and Suggestions for Authors

The work covers an important issue of the development of the professional practice of pharmacists.  I make no comments on the content. Relevant research tools were used to this kind of review article. The work needs refinement on the editorial side. The supplementary materials indicate that they include: "Figure 1 (PRISMA Flow-chart), Supplementary S1 (Search Strategy), Table 1 (Characteristics Table), Table 2 (Outcomes Table)".

However, in the file (supplementary materials) there is only Table 1.

According to the content in the article, I suggest moving from the content of the article : Table 1 (Characteristics Table), Table 2 (Outcomes Table) to supplementary materials also.  I do not suggest any additional comments. 

Reviewer 2 Report

Comments and Suggestions for Authors

1 - This paper provides a scoping review of 10 studies to expand community pharmacists’ scope of practice. The sites of the study were international. The period of the sites studied was 2013-2023, and the study was required to be a randomized clinical trial.

2 - The strengths of this study are investigating the premise that community pharmacists can increase access for many groups of persons beyond primary care clinics, particularly in rural areas.  Cost and ease of access are also important factors that make this concept important. The paper does an excellent job of detailing the 10 RCT sites.

3 - The weakness of the paper is that the sites had very different focuses on what they were trying to investigate and achieve.  I would like to have more information about each site. How many employees; what is the size of the pharmacy; where is it located related to other healthcare facilities; how is a site different than others; what is the volume of prescriptions handled and how many customers? What were some of the negatives? Was any harm done? How are pharmacists certified or were they? What are the checks and balances?  How was inappropriate prescribing to sell more medications prevented or checked?  Were there conflicts of interest? The paper and the studies were looking at only a few of the outcomes or the important questions. What motivated the study? The paper focuses on determining if a positive change in the system occurred.  I think there are better ways to evaluate the project.  What were the before and after in several areas?

       What is the relationship between the pharmacists and the physicians? For example, Are the pharmacist’s actions transmitted back to the MD for inclusion in the EHR? Are there communication paths between MD and pharmacist?

4 – You need a better way of bringing the studies together. I would like to see a conclusion that took what you learned from the 10 studies to present what the scope of practice might be.  If possible, I would like to see some interviews of the patients and what they were told, how they were motivated, and why they followed or did not follow the recommendations I would like the style of the paper changed to reflect more of what positives you learned across the 10 studies, rather than only discussing each site individually.

       I do like this paper, and I think it has value as a stimulator for what others might try.  I would like to see a more consolidated study across sites in which they all used the same methods and objectives.

5 -I had some difficulty tracking the data in Tables 1 and 2. My suggestion is to put the legend on the top of every page.  Second, I would suggest that you make the transition to a new site more obvious.  I would like to see each new site start a new page.

Reviewer 3 Report

Comments and Suggestions for Authors

A study using the Scoping Review Framework as guided by the Joanna Briggs Institute was developed with the objective to describe interventions that expand the pharmacist's scope of practice within a community pharmacy setting and assess where there has been a change to the scope of pharmacy practice by those interventions, and to determine their effectiveness.

The relevant topic of the role of community pharmacies and their expansion was addressed in the search for evidence of impact. The introduction provides a good contextualization of the problem and the approach used

The methodology meets the requirements listed. The tables are well organized and explicit, the validation mechanisms have been complied with. A mapping was carried out considering the themes of the Pharmaceutical Society of Australia Professional Practice Standards 2023

The analysis of limitations and strengths is realistic and the conclusions are aligned with what has been achieved. At a time of important changes in health and its actors, it is important that decisions can be evidence based
